# Heavy Ion-Responsive lncRNA EBLN3P Functions in the Radiosensitization of Non-Small Cell Lung Cancer Cells Mediated by TNPO1

**DOI:** 10.3390/cancers15020511

**Published:** 2023-01-13

**Authors:** Haoyi Tang, Hao Huang, Zi Guo, Haitong Huang, Zihe Niu, Yi Ji, Yuyang Zhang, Huahui Bian, Wentao Hu

**Affiliations:** 1State Key Laboratory of Radiation Medicine and Protection, School of Radiation Medicine and Protection, Collaborative Innovation Center of Radiological Medicine of Jiangsu Higher Education Institutions, Soochow University, Suzhou 215123, China; 2Nuclear and Radiation Incident Medical Emergency Office, The Second Affiliated Hospital of Soochow University, Suzhou 215004, China

**Keywords:** non-small cell lung cancer, carbon ion radiotherapy, lncRNA EBLN3P, miR-144-3p, TNPO1, ceRNA

## Abstract

**Simple Summary:**

Heavy-ion radiotherapy (HIRT) is associated with higher tumor cure rates compared with conventional radiotherapy (CRT). However, considering the high cost of HIRT, most tumor patients are still treated with CRT. The aim of this study was to elucidate the tumor inhibitory mechanism of HIRT by exploring gene expression signatures after heavy-ion exposure. We confirmed that the carbon ion-responsive long non-coding RNA endogenous bornavirus-like nucleoprotein 3, pseudogene (EBLN3P), is significantly decreased in carbon-ion irradiated non-small cell lung cancer (NSCLC) cells. The combination therapy of LNC EBLN3P-inhibition and X-ray irradiation can delay the progression of NSCLC both in vitro and in vivo, indicating the potential role of LNC EBLN3P as a target of radiosensitization in CRT.

**Abstract:**

In recent decades, the rapid development of radiotherapy has dramatically increased the cure rate of malignant tumors. Heavy-ion radiotherapy, which is characterized by the “Bragg Peak” because of its excellent physical properties, induces extensive unrepairable DNA damage in tumor tissues, while normal tissues in the path of ion beams suffer less damage. However, there are few prognostic molecular biomarkers that can be used to assess the efficacy of heavy ion radiotherapy. In this study, we focus on non-small cell lung cancer (NSCLC) radiotherapy and use RNA sequencing and bioinformatic analysis to investigate the gene expression profiles of A549 cells exposed to X-ray or carbon ion irradiation to screen the key genes involved in the stronger tumor-killing effect induced by carbon ions. The potential ceRNA network was predicted and verified by polymerase chain amplification, western blotting analysis, colony formation assay, and apoptosis assay. The results of the experiments indicated that lncRNA EBLN3P plays a critical role in inhibiting carbon ion-induced cell proliferation and inducing apoptosis of NSCLC cells. These functions were achieved by the EBLN3P/miR-144-3p/TNPO1 (transportin-1) ceRNA network. In summary, the lncRNA EBLN3P functions as a ceRNA to mediate lung cancer inhibition induced by carbon ion irradiation by sponging miR-144-3p to regulate TNPO1 expression, indicating that EBLN3P may be a promising target for increasing the treatment efficacy of conventional radiotherapy for NSCLC.

## 1. Introduction

According to the most recent annual cancer statistics report, lung cancer is the most prevalent type of malignancy leading to cancer-related deaths worldwide, with an estimated annual death toll of 1.8 million [1]. Among lung cancer patients, approximately 85% of cases are characterized as non-small cell lung cancer (NSCLC), which is the main pathological type [2]. Despite significant improvements in the currently available treatment strategies, only 22% of lung cancer patients survive for more than five years in the United States [3]. In the last three decades, heavy ion radiotherapy, the efficacy of which is better than that of conventional photon radiotherapy, has been demonstrated to be beneficial in the treatment of NSCLC. From 1994 to 1999, the scholars in National Institute of Radiological Science in Japan used carbon ions to treat 81 cases of stage I NSCLC in phase I and II clinical trials, with local control rate of 64% in phase I trial and 84% in phase II trial, and the 5-year overall survival rate of 81 patients was 42%, which were better than the effects of conventional radiotherapy and similar to the effects of surgery [4]. However, the mechanism of action of heavy ions in the suppression of tumor growth is unclear, which limits its optimization. Therefore, delineating the mechanisms underlying the differential response of lung cancer cells to photons and carbon ions is of great significance for improving the treatment efficacy.

Long non-coding RNAs (lncRNAs) are transcripts of more than 200 nucleotides that do not encode proteins. They have been demonstrated to participate in many biological processes, including the regulation of genetic transcription and cell proliferation [5]. Their aberrant expression or dysfunction can lead to the development of numerous diseases such as cancer [5], and there is increasing evidence for the roles of lncRNAs in tumor development and progression via a ceRNA mechanism [6,7,8]. However, there is no study on the differential response of lncRNAs to photon and particle irradiation. To clarify the mechanisms underlying the differential response of cancer cells to photon and heavy ion irradiation, we performed RNA sequencing on NSCLC A549 cells exposed to X-rays and carbon ions and found that carbon ion irradiation induced a significant decrease in the expression of lncRNA EBLN3P (LNC EBLN3P) compared to X-rays in A549 cells. According to bioinformatics predictions, down-regulated LNC EBLN3P may play an essential role in the carbon ion radiation-induced suppression of lung cancer by modulating the oncogene TNPO1, which is mediated by miR-144-3p.

In this study, we explored the biological mechanisms underlying carbon ion radiotherapy in NSCLC. We identified LNC EBLN3P through RNA-sequencing (RNA-seq), a carbon ion-responsive lncRNA that can regulate A549 cell apoptosis. Through mechanistic experiments, we demonstrated that LNC EBLN3P/miR-144-3p/TNPO1 forms a ceRNA network and plays a crucial role in carbon ion-induced tumor growth inhibition. In conclusion, from an unconventional perspective, our findings revealed a potential mechanism underlying the effectiveness of carbon ion radiotherapy in the treatment of NSCLC.

## 2. Materials and Methods

### 2.1. Cell Culture and Irradiation

BEAS-2B (human lung bronchial epithelial cells), A549, H1299, HCC827, and Calu-1 (NSCLC cells) were purchased from the National Collection of Authenticated Cell Cultures (Shanghai, China). Dulbecco’s modified Eagle medium (Thermo Fisher Scientific, Waltham, MA, USA) was used for culturing BEAS-2B cells, whereas Roswell Park Memorial Institute-1640 medium (Thermo Fisher Scientific) was used for culturing lung cancer cells. All media were supplemented with 1% penicillin–streptomycin and 10% fetal bovine serum (FBS). Cells were cultured in an incubator containing 5% CO_2_ at 37 °C. All cell lines were cultured, maintained, and used in the range of 10 to 20 passages. The RS 2000 X-ray Biological Irradiator (Rad Source Technologies, Suwanee, GA, USA) was used to produce X-rays (225 kVp, 1.12 Gy/min). The Heavy Ion Medical Accelerator in Chiba (HIMAC) at the National Institute of Radiological Science was used to produce carbon ion beams (290 MeV/u, 1.22 Gy/min). The linear energy transfer (LET) of C290 at the entrance of the plateau was 13.3 keV/μm, whereas the LET in the spread-out Bragg peak was 80 keV/μm.

### 2.2. Cell Transfection

A549 cells were seeded in 6-well plates and cultured until 60% confluent. Cells were transfected with LNC EBLN3P small interfering RNA (si-LNC EBLN3P), negative control small interfering RNA (si-NC), LNC EBLN3P overexpression plasmid (pcDNA3.1-LNC EBLN3P), empty vector (pcDNA3.1), miR-144-3p inhibitor (miR-inhibitor), and negative control miR inhibitor (miR-NC). All siRNAs and plasmids were synthesized by Sangon Biotech (Shanghai, China), and sequences used to prepare siRNAs and plasmids are listed in Table 1.

### 2.3. qRT-PCR

TRIzol reagent (Thermo Fisher Scientific) was used to extract total RNA from cells and tissues. The Prime Script RT kit (Takara Shuzo Co., Shiga, Japan) was used for the reverse transcription of mRNA. For the polymerase chain reaction, the AceQ qPCR SYBR Green Master Mix kit (low ROX premixed) (Vazyme Biotech Co., Ltd., Nanjing, China) was used. The Vii7A system (Thermo Fisher Scientific) was used for quantitative PCR and signal generation. The comparative C(t) method was employed for data analysis. Glyceraldehyde-3-phosphate dehydrogenase was used as the internal reference gene, and sequences used for PCR are listed in Table 2. The primers for the reverse transcription and amplification of miR-144-3p were designed and synthesized by RiboBio (Guangzhou, China).

### 2.4. Western Blotting

Radioimmunoprecipitation assay lysis buffer (Beyotime, Shanghai, China) was used to extract total protein from cells, and the protein concentration of each sample was determined using the BCA Protein Assay kit (Beyotime, Shanghai, China). Equivalent amounts of protein were separated by sodium dodecyl sulfate polyacrylamide gel electrophoresis and subsequently transferred to a polyvinylidene fluoride membrane (Amersham, Arlington Heights, IL, USA). The chemiluminescence signals were developed using an Enhanced Chemiluminescence Detection kit (Millipore, Burlington, MA, USA) and acquired using a polychromatic fluorescence chemiluminescence imaging analysis system (Alpha Innotech, San Leandro, CA, USA). Antibodies against transportin-1 (catalog no. 38700; Cell Signaling Technology, Danvers, MA, USA) and glyceraldehyde-3-phosphate dehydrogenase (catalog no. 5174; Cell Signaling Technology) were used for western blotting.

### 2.5. Cell Viability

The Cell Counting Kit-8 assay (Dojindo, Kumamoto, Japan) was used to assess cell viability. Cells were digested with trypsin and seeded in 96-well plates at a density of 2000 cells per well. The medium was replaced with fresh medium containing 10% CCK-8 reagent, and cells were cultured in an incubator at 37 °C for 2 h in the dark. Thereafter, the optical density (OD) value of each well was measured at 450 nm with a multifunctional microplate reader (BioTek Instruments, Winooski, VT, USA). To determine the number of viable cells, the cell number was proportional to the OD value.

### 2.6. Apoptosis

A549 cells were digested with trypsin, seeded in 6-well plates at a density of 1 × 10^5^ per well, and collected at 24 h after transfection or irradiation. Cells were stained with propidium iodide and Annexin-V according to the instructions of the Annexin V-Alexa Fluor 647–Propidium Iodide Apoptosis Detection kit (FCMACS Biotech, Nanjing, China), and the apoptotic rate was determined by flow cytometry (BD Biosciences, Franklin Lakes, NJ, USA).

### 2.7. Colony Formation

Cells were seeded in T25 plastic flasks in triplicate before irradiation. Cells were exposed to a single dose of 0-, 1-, 2-, 4-, 6-Gy X-ray irradiation, and the X-ray beams entered cells from the bottom of the bottle. After irradiation, the cells were counted, plated into Φ60 mm dishes and incubated for 14 days, after which they were fixed with 75% ethanol for 15 min at room temperature and stained with 0.1% crystal violet for 20 min. The survival fraction (SF) of cells at each irradiation dose was determined by the following formula:SF = (no. colonies formed)/(no. colonies seeded) × 100%.(1)

### 2.8. Luciferase Assay

A549 cells were seeded in 6-well plates and co-transfected with the luciferase reporter plasmid LNC EBLN3P 3′UTR wild-type (WT) or 3′UTR mutant (MUT) and the miR-144-3p mimic or negative control (miR-NC). After 48 h of transfection, the cells were processed according to the instructions of the Dual-Luciferase Reporter Gene Assay kit (Beyotime, Shanghai, China), and Firefly and Renilla luciferase activities were measured with a multifunctional microplate reader (BioTek Instruments). All vectors were synthesized by RiboBio.

### 2.9. Animal Treatment

Male BALB/c nude mice at 6–8 weeks of age were purchased from SLACCAS Animal Laboratory (Shanghai, China) and housed under specific-pathogen-free conditions. A549 cells (5 × 10^6^) were subcutaneously injected into the right flanks of the mice. After two weeks, when the tumor volume reached approximately 70 mm^3^, the tumors were exposed to a single dose of 8-Gy X-ray irradiation using a cone-beam CT-guided precision irradiation system (X-RAD 225Cx; Precision X-Ray, North Branford, CT, USA). Thereafter, si-LNC EBLN3P or si-NC (RiboBio, Guangzhou, China) was injected into the solid tumors every 4 days at a dose of 1 nmol (dissolved in 25 μL sterile PBS) each tumor. The tumor size was measured with a caliper every 4 days for 16 days, and the volume was determined by the following formula:V = ab^2^ × 0.52
where a indicates the length and b indicates the width. After 16 days, the mice were sacrificed, and tumors were harvested and weighed. All animal experiments complied with ethical guidelines, and the Soochow University Institutional Animal Care and Use Committee approved all animal experiments (ECSU-2021000109).

### 2.10. Hematoxylin–Eosin Staining and Immunohistochemistry

Tissues were embedded in paraffin, and tissue blocks were sectioned after gradient dehydration. Sections were stained with hematoxylin–eosin according to the instructions of the Hematoxylin-Eosin Stain kit (Solarbio, Beijing, China). For immunohistochemistry, sections were incubated with TNPO1 (catalog no. ab10303; Abcam, Cambridge, UK) and BAX (catalog no. 5023; Cell Signaling Technology) antibodies overnight at 4 °C. The next day, the sections were washed with phosphate-buffered saline containing Tween-20 and incubated with a secondary antibody (catalog nos. PV6001, PV6002; ZSGB-Bio, Beijing, China) at 37 °C for 30 min. Sections were stained with 3,3′-diaminobenzidine, dehydrated, and sealed with neutral balsam. Images were acquired with an inverted microscope (Leica, Wetzlar, Germany), and the cells positively stained with TNPO1 and BAX were analyzed with IHC Profiler (an Image-J plugin).

### 2.11. Statistical Analysis

GraphPad Prism 8.0 or SPSS 16.0 software packages were used for the statistical analysis of the data. There were at least three biological replicates in each experiment. Student’s *t*-test was used for the comparison of means of two groups and ANOVA was used for the comparison of means of three or more groups. Data are presented as mean ± standard deviation (SD). Statistical differences were considered significant at *p* < 0.05.

## 3. Results

### 3.1. Differential Expression of lncRNAs after Carbon Ion and X-ray Irradiation

To identify the ionizing radiation responsive lncRNA signatures, an expression profile of lncRNAs in irradiated A549 cells was investigated. RNA was isolated from A549 cells 2 h after exposed to X-ray (2 Gy) or carbon ion (2 Gy) irradiation and subjected to RNA-seq. According to mRNA/lncRNA/miRNA interaction analysis, we constructed a ceRNA regulatory network reflecting the differential response to the two kinds of radiation, among which LNC EBLN3P was demonstrated to play a significant role by interacting with miR-144-3p and TNPO1 (Figure 1A). The number of reported AGO-CLIP experiments also implied that TNPO1 was the most reliable target of miR-144-3p among the 6 differentially expressed candidate targets (Appendix A). LNC EBLN3P expression was reduced by carbon ion irradiation, which is less significantly down-regulated by X-ray irradiation (Figure 1B). An analysis of the data from gene expression profiling interactive analysis (GEPIA) revealed that the expression of LNC EBLN3P was slightly higher in lung cancer tissues; however, no significant difference was noted in LNC EBLN3P expression level between tumor and normal tissues or in survival rate between high and low LNC EBLN3P patients (Figure 1C,D), and lung cancer tissues have increased expression of TNPO1 compared with paired normal tissues (Figure 1F). Furthermore, in samples of lung cancer patients from GEPIA2 (http://gepia2.cancer-pku.cn/#survival/ (accessed on 26 November 2022)), we observed that patients with high TNPO1 expression had lower overall survival (OS) than those with low expression of TNPO1 (Figure 1G). An analysis of the co-expression results indicated a positive correlation between LNC EBLN3P and TNPO1 in both lung adenocarcinoma and lung squamous carcinoma (Figure 1E,H). These results show that LNC EBLN3P is downregulated after irradiation, suggesting that radiation-responsive LNC EBLN3P may play a role in the radiosensitivity of lung cancer cells.

### 3.2. Irradiation Causes Expression Changes of LNC EBLN3P

To determine whether LNC EBLN3P is expressed differently in lung cancer cells and normal lung cells, we performed qRT-PCR experiments using normal BEAS-2B cells and four NSCLC cell lines (A549, H1299, HCC827, and Calu-1) for validation. As shown in Figure 2A, the basal expression level of LNC EBLN3P was higher in lung cancer cells compared with BEAS-2B cells. We further examined TNPO1 expression, and the results showed that the TNPO1 protein level was also higher in A549 and H1299 cells compared with BEAS-2B cells (Figure 2B, Appendix A).

To explore whether the expression of LNC EBLN3P and TNPO1 responds to irradiation, we investigated the mRNA levels of LNC EBLN3P and TNPO1 in response to different types of radiation. Total RNA was isolated from A549 cells 6, 12, 18, or 24 h after exposed to X-ray or carbon ion irradiation, followed by qRT-PCR experiments. As shown in Figure 2C,D, the transcription levels of LNC EBLN3P and TNPO1 were both up-regulated at all time points, and the down-regulation in expression induced by carbon ion beams was more significant than that induced by X-rays. Next, we examined the expression of LNC EBLN3P in A549 cells exposed to different doses of X-ray irradiation. The results showed that LNC EBLN3P expression was down-regulated with increasing radiation dose (Figure 2E). Moreover, the reduction in the TNPO1 protein level by 2-Gy carbon ion irradiation was greater than that by 2-Gy X-ray irradiation (Figure 2F, Appendix A). These results provide further proof that radiation can decrease the expression of LNC EBLN3P and TNPO1 in lung cancer cells.

### 3.3. LNC EBLN3P Reduces the Viability of A549 Cells

To verify whether LNC EBLN3P could affect the viability of cells, we transfected A549 cells with pcDNA3.1-LNC EBLN3P or si-LNC EBLN3P to up-regulate or down-regulate the expression of LNC EBLN3P (Figure 3A,C). Cell proliferation was enhanced at 48 h and significantly enhanced at 96 h by the up-regulation of LNC EBLN3P expression (Figure 3B), whereas the reduction of cell viability induced by the down-regulation of LNC EBLN3P expression persisted from 24 h to 96 h (Figure 3D). We further explored the viability and radiosensitivity of A549 cells after overexpression or knockdown of LNC EBLN3P. We found that the overexpression of LNC EBLN3P significantly decreased the apoptotic rate (Figure 3E,F), whereas the knockdown of LNC EBLN3P significantly increased the apoptotic rate after 2-Gy X-ray irradiation (Figure 3G,H). As for radiosensitivity, the overexpression of LNC EBLN3P increased colony formation, whereas the knockdown of LNC EBLN3P decreased colony formation after X-ray irradiation (Figure 3I,J). Taken collectively, these results indicate that the down-regulation of LNC EBLN3P expression can decrease the viability and increase the radiosensitivity of A549 cells.

### 3.4. LNC EBLN3P Regulates the Expression of TNPO1

As an oncogene, TNPO1 has been reported to be associated with tumor growth, invasion, and metastasis [9,10,11]. To determine whether LNC EBLN3P can regulate TNPO1 expression in lung cancer cells, qRT-PCR and western blotting experiments were performed to examine the TNPO1 mRNA and protein levels in LNC EBLN3P-overexpressed or LNC EBLN3P-silenced A549 cells. As shown in Figure 4A, the expression of TNPO1 in LNC EBLN3P-overexpressed A549 cells was more than 15-fold higher than that in control cells, and the TNPO1 protein level also showed a significant increase (Figure 4C, Appendix A). By contrast, both TNPO1 mRNA and protein levels were lower in LNC EBLN3P-silenced A549 cells than that in control cells (Figure 4B,D, Appendix A). Moreover, it was found that TNPO1-knockdown inhibited proliferation of A549 cells and sensitized the cells to X-ray irradiation by detection of the proliferation, apoptosis, and survival of cells as shown in Appendix A. These results indicate that LNC EBLN3P positively regulates TNPO1 expression in NSCLC cells.

### 3.5. MiR-144-3p Mediates the Regulation of LNC EBLN3P on TNPO1

Based on bioinformatics analysis, we predicted the potential ceRNA network and observed that the regulation of LNC EBLN3P on TNPO1 was mediated by miR-144-3p (Figure 1A). To verify the interaction between miR-144-3p and LNC EBLN3P or TNPO1, we co-transfected A549 cells with luciferase reporter plasmids containing LNC EBLN3P 3′UTR wild-type (WT) or 3′UTR mutant (MUT) and miR-144-3p mimic or negative control (miR-NC) and measured the relative luciferase activity in cells. The results showed that the luciferase activity of the LNC EBLN3P WT and miR-144-3p co-transfected group significantly decreased, which confirmed the predicted binding site between miR-144-3p and LNC EBLN3P (Figure 5A). Similar results were obtained by the luciferase-reporter assay, which confirmed the predicted binding site between miR-144-3p and TNPO1 (Figure 5B). qRT-PCR detection suggested that miR-144-3p was upregulated by both X-ray and carbon-ion irradiation. However, the upregulation induced by carbon ion beams was more significant than that induced by X-rays (Figure 5C). Next, we co-transfected A549 cells with LNC EBLN3P siRNA (si-LNC EBLN3P) and miR-144-3p inhibitor (miR-inhibitor) and then detected the effects on cell proliferation, apoptosis, and survival. Our results showed that LNC EBLN3P knockdown decreased TNPO1 mRNA expression and suppressed cell viability, while increased cellular apoptosis and radiosensitivity. All of these effects could be rescued by knockdown of miR-144-3p using its inhibitors (Figure 5D–H). Moreover, the inhibition of LNC EBLN3P induced miR-144-3p upregulation, which was curtailed by miR-inhibitor transfection (Appendix A). Taken collectively, these results indicate that the regulation of LNC EBLN3P on TNPO1 was mediated by miR-144-3p, and the LNC EBLN3P/miR-144-3p/TNPO1 axis, which was inactivated after irradiation, plays a role in the death of NSCLC cells.

### 3.6. Inhibition of LNC EBLN3P Radiosensitizes NSCLC Cells In Vivo through the miR-144-3p/TNPO1 Axis

To explore the potential of LNC EBLN3P in the radiosensitization of photon radiotherapy-induced NSCLC inhibition, we irradiated the tumors with 8-Gy X-rays using a cone-beam CT-guided precision irradiation system. After two weeks, the mice were sacrificed and the pathological changes and expression of LNC EBLN3P and TNPO1 in lung tumor tissues were examined. Compared with the X-ray irradiation group, X-ray irradiation combined with LNC EBLN3P knockdown significantly suppressed tumor development (Figure 6A–C). After X-ray irradiation, the knockdown of LNC EBLN3P significantly decreased the protein expression of TNPO1 in lung cancer tissues compared with the control (Figure 6D,E). Furthermore, the results of BAX expression experiments and hematoxylin–eosin staining showed that knockdown of LNC EBLN3P induced necrosis and apoptosis in lung cancer tissues (Figure 6D,F). Next, total RNA was isolated from tissues for qRT-PCR, and the results indicated that the si-LNC EBLN3P treatment group exhibited much lower expression levels of LNC EBLN3P and TNPO1 than the si-NC group (Figure 6H, I). These findings indicate that LNC EBLN3P knockdown increased the radiosensitivity of NSCLC cells through the miR-144-3p/TNPO1 axis.

## 4. Discussion

Radiotherapy has a long history in the treatment of lung cancer, but conventional photon beams, such as γ-rays and X-rays, have many disadvantages, including low efficacy rates in radiation-resistant tumors, the tendency to induce varying degrees of radiation damage, and high tumor recurrence rates. The advent of carbon ion therapy has revolutionized radiotherapy. To date, several clinical studies have shown that carbon ion radiotherapy has excellent efficacy for different stages of lung cancer [12,13]. However, its mechanism of action in NSCLC is unclear, thereby presenting many obstacles to further improving its efficacy and reducing its side effects.

In this study, we found that LNC EBLN3P expression in A549 cells was downregulated by carbon ion irradiation, which was less significantly down-regulated by X-ray irradiation, indicating EBLN3P may play an important role in carbon ion-induced lung cancer cell death. Indeed, knockdown of EBLN3P inhibited cell proliferation and colony formation and promoted A549 cell apoptosis. Our data further revealed that LNC EBLN3P positively regulated the expression of TNPO1, an oncogene. The results of luciferase assays revealed an interaction between miR-144-3p and LNC EBLN3P or TNPO1, and further analysis demonstrated that TNPO1 can interact with miR-144-3p and its expression was positively regulated by LNC EBLN3P. Down-regulation of LNC EBLN3P expression caused the upregulation of miR-144-3p expression, which in turn caused the level of TNPO1 to increase, thereby inhibiting the viability and enhancing the radiosensitivity of lung cancer cells. Thus, our study demonstrates that carbon ion-responsive LNC EBLN3P promotes TNPO1 expression by sponging miR-144-3p and LNC EBLN3P/miR-144-3p/TNPO1 forms a ceRNA network.

Recent studies have demonstrated that lncRNAs are crucial regulators of tumor growth and invasion; they have reported that LNC EBLN3P can function as an oncogene in osteosarcoma and lung adenocarcinoma, and the inhibition of EBLN3P can be used in the treatment of cancer [14,15]. LNC EBLN3P can also promote the recovery of impaired spiral ganglion neurons by regulating the miR-204-5p/TMPRSS3 axis [16]. Although there is no universally accepted set of specific lncRNA markers for diagnosing NSCLC, numerous studies have investigated aberrantly and differentially expressed lncRNAs [17,18], and multiple lncRNAs may be used in the future as diagnostic markers for the pathological staging of NSCLC. In terms of treatment, drug resistance in NSCLC patients is often the main reason for treatment failure. The lncRNAs NNT-AS1 and HOXA-AS3 have been reported to associate with cisplatin resistance, which is the standard adjuvant treatment after chemotherapy for advanced NSCLC [19,20]. In addition, the effect of lncRNA on epidermal growth factor receptor-tyrosine kinase inhibitor (EGFR-TKI) resistance is also a widely investigated research area [21]. The roles of lncRNAs in the efficacy of radiotherapy in non-surgically treated patients with advanced NSCLC are also gradually attracting attention. Wang et al. reported that the expression of the lncRNA CCAT1, which is highly expressed in NSCLC cells, was down-regulated after irradiation, and its knockdown inhibited the mitogen-activated protein kinase pathway, which increased the radiosensitivity of NSCLC cells [22], while Ma et al. demonstrated that the lncRNA LINC00460 can promote gefitinib resistance in NSCLC cells by sponging miR-769-5p to modulate EGFR [23]. The radiation resistance of lung cancer cells has long been a source of frustration for the field of radiotherapy. Unlike most studies that only investigate the role of X-rays in tumor cell sensitization, our study focused on the epigenetic difference in the radiation response between X-rays and heavy ions and examined differentially expressed lncRNAs. Through bioinformatics analyses and mechanistic studies, we identified LNC EBLN3P as a potential molecule functioning in the response of tumor cells to heavy ions through a ceRNA regulatory network.

A wealth of experimental evidence suggests that lncRNAs and miRNAs have binding sites to co-regulate the expression of target genes. Previous studies have attempted to elucidate the mechanistic pathways of lncRNAs in NSCLC through a ceRNA perspective. For example, Zhang et al. reported that MAGI1-IT1 could stimulate the NSCLC cell proliferation and growth by up-regulating AKT1 as a ceRNA [24]. Wang et al. demonstrated that LINC01234 promoted the progression of NSCLC through the miR-433-3p/GRB2 axis [25]. However, up to now, no ceRNA regulatory network has been identified to be involved in the radiobiological effects of heavy ions. We identify, for the first time, a ceRNA network functioning in the regulation of heavy ion-induced NSCLC cell killing and confirm the regulation of LNC EBLN3P on miR-144-3p, which can target and inhibit TNPO1 expression. The results revealed that LNC EBLN3P knockdown decreased cell proliferation, whereas the miR-144-3p antagonist increased cell proliferation. In other words, the miR-144-3p antagonist could rescue the knockdown effects of LNC EBLN3P, thereby allowing normal cell proliferation, which is indicative of their direct regulatory relationship. In addition, TNPO1 is identified as an oncogene and has been reported to be associated with the development of esophageal cancer and cervical cancer [9,26], consistent with our finding that TNPO1-knockdown inhibited proliferation of A549 cells. In our study, we reveal that the knockdown of LNC EBLN3P induced by carbon ion irradiation could inhibit cell proliferation and induce apoptosis and radiosensitivity of NSCLC cells through the down-regulation of TNPO1, demonstrating that TNPO1 may be a key factor functioning in the radiobiological effects of heavy ions.

## 5. Conclusions

In conclusion, we report that the LNC EBLN3P/miR-144-3p/TNPO1 axis responds to carbon ion-induced apoptosis of NSCLC cells in vitro and in vivo, implying this pathway may hold promise for improving the treatment efficiency of radiotherapy for NSCLC.

## Figures and Tables

**Figure 1 cancers-15-00511-f001:**
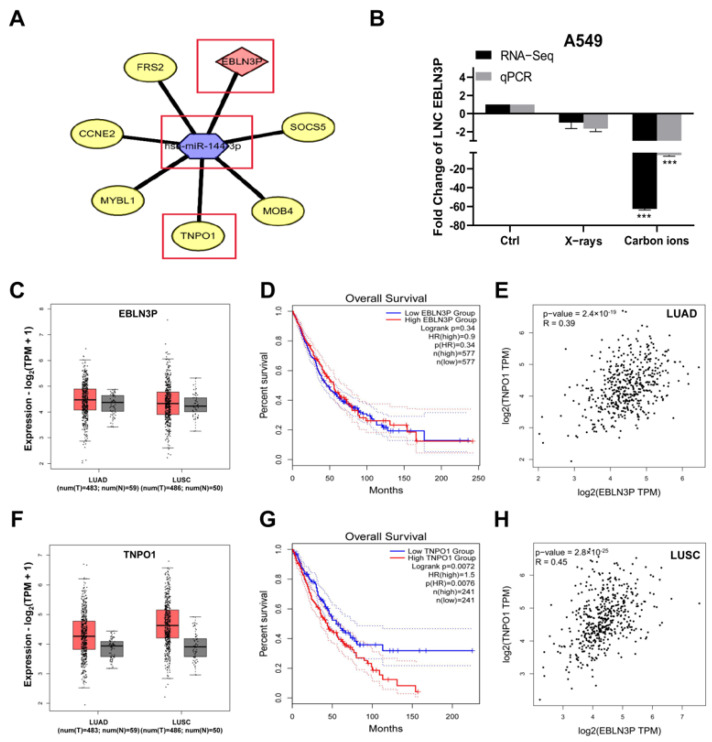
LNC EBLN3P expression is down−regulated by X−ray or carbon ion irradiation in A549 cells. (**A**) The potential ceRNA network is predicted according to the highest down−regulated lncRNA, EBLN3P, which is also named as LNC EBLN3P. (**B**) The relative expression of LNC EBLN3P in A549 cells after X−ray (2 Gy) or carbon ion (2 Gy) irradiation was analyzed by RNA−seq and qRT−PCR. (**C**,**F**) The relative expression of LNC EBLN3P and TNPO1 in lung cancer tissues (T) and normal tissues (N). (**D**,**G**) The overall survival (OS) of lung cancer patients with high expression (red line) and low expression (blue line) of LNC EBLN3P or TNPO1. (**E**,**H**) The co−expression analysis of EBLN3P and TNPO1 in lung adenocarcinoma (LUAD) and lung squamous carcinoma (LUSC). *** *p* < 0.001.

**Figure 2 cancers-15-00511-f002:**
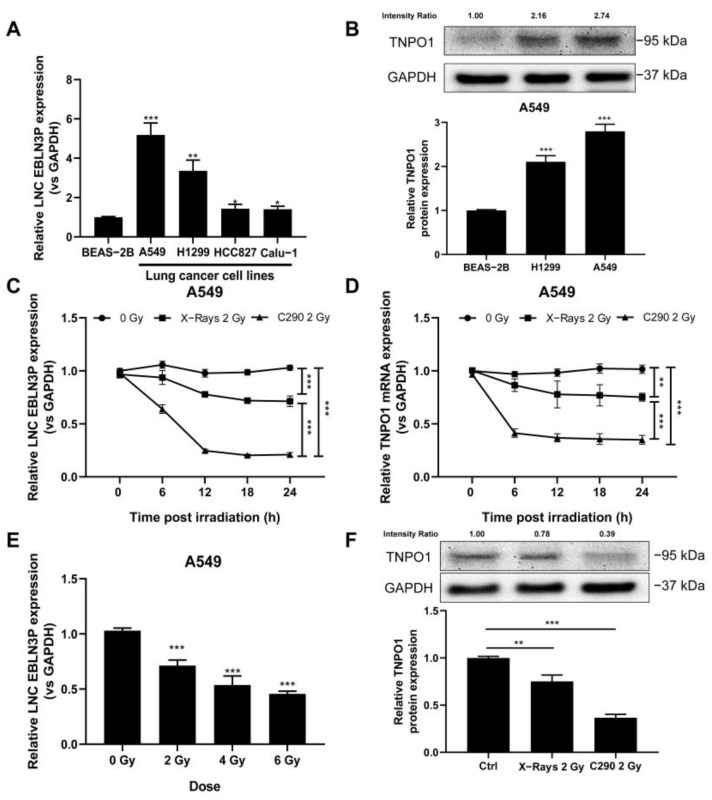
X−ray and carbon ion irradiation induced differential downregulation of LNC EBLN3P in lung cancer cells. (**A**) The basic expression levels of LNC EBLN3P in BEAS−2B and NSCLC cells (A549, H1299, HCC827 and Calu−1) were tested by qRT−PCR. (**B**) The TNPO1 protein levels of cells were tested by western blotting. (**C**,**D**) The mRNA levels of LNC EBLN3P and TNPO1 are both down−regulated over time in A549 cells irradiated with X−rays or carbon ions. (**E**) Expression of LNC EBLN3P was tested in A549 cells which suffered different doses of X−ray irradiation (samples were collected 24 h after irradiation). (**F**) TNPO1 protein levels were determined with western blotting in A549 cells exposed to the same dose of X−rays and carbon ions. Data shown in A–E represent the mean ± SD (*n* = 3). * *p* < 0.05, ** *p* < 0.01, *** *p* < 0.001.

**Figure 3 cancers-15-00511-f003:**
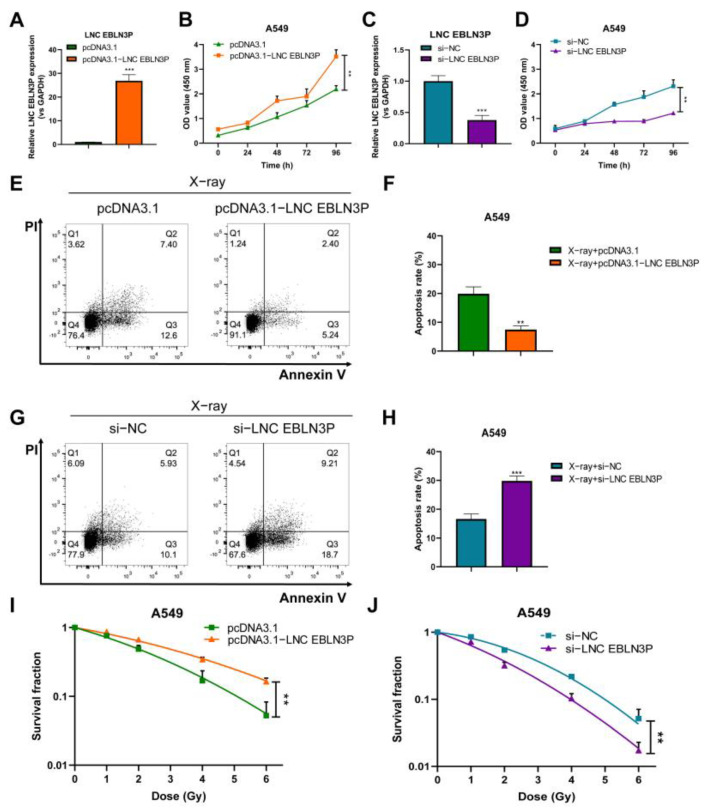
Downregulation of LNC EBLN3P inhibits cell proliferation and colony formation and enhances cell apoptosis. (**A**–**D**) The overexpression of LNC EBLN3P increases the proliferation of A549 cells, whereas its knockdown decreases the viability of cells. (**E**–**J**) The overexpression of LNC EBLN3P decreases both the apoptotic rate and radiosensitivity of cells, whereas its knockdown increases the apoptotic rate and radiosensitivity. Data shown in (**A**–**J**) represent the mean ± SD (*n* = 3). ** *p* < 0.01, *** *p* < 0.001.

**Figure 4 cancers-15-00511-f004:**
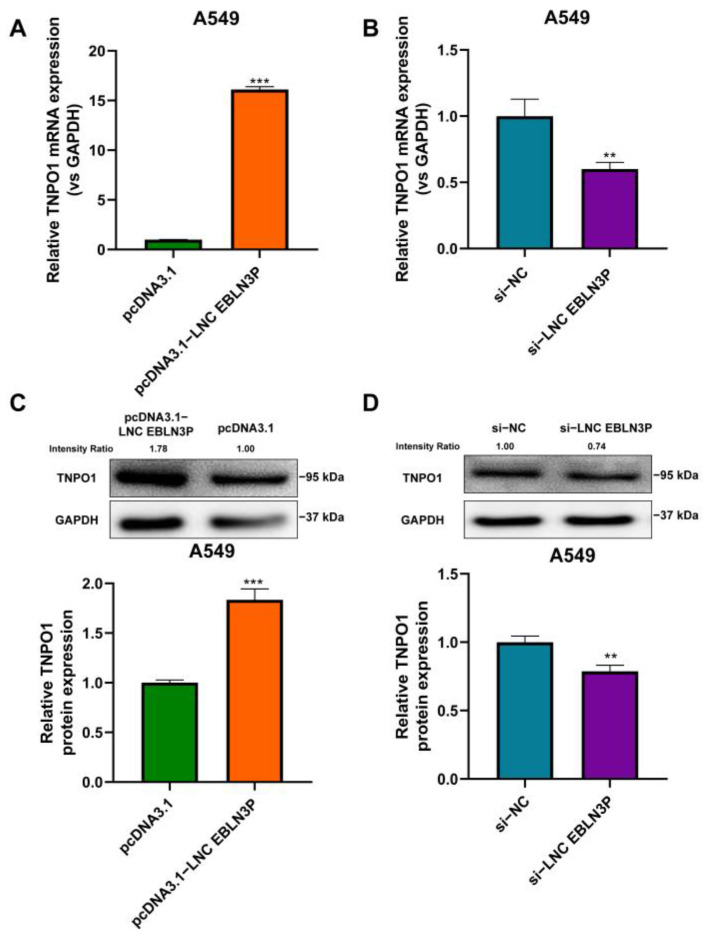
LNC EBLN3P positively regulates TNPO1 expression in lung cancer cells. (**A**,**C**) The overexpression of LNC EBLN3P increases both TNPO1 mRNA and protein levels in A549 cells. (**B**,**D**) The knockdown of LNC EBLN3P decreases both TNPO1 mRNA and protein levels in A549 cells. Data shown in A–D represent the mean ± SD (*n* = 3). ** *p* < 0.01, *** *p* < 0.001.

**Figure 5 cancers-15-00511-f005:**
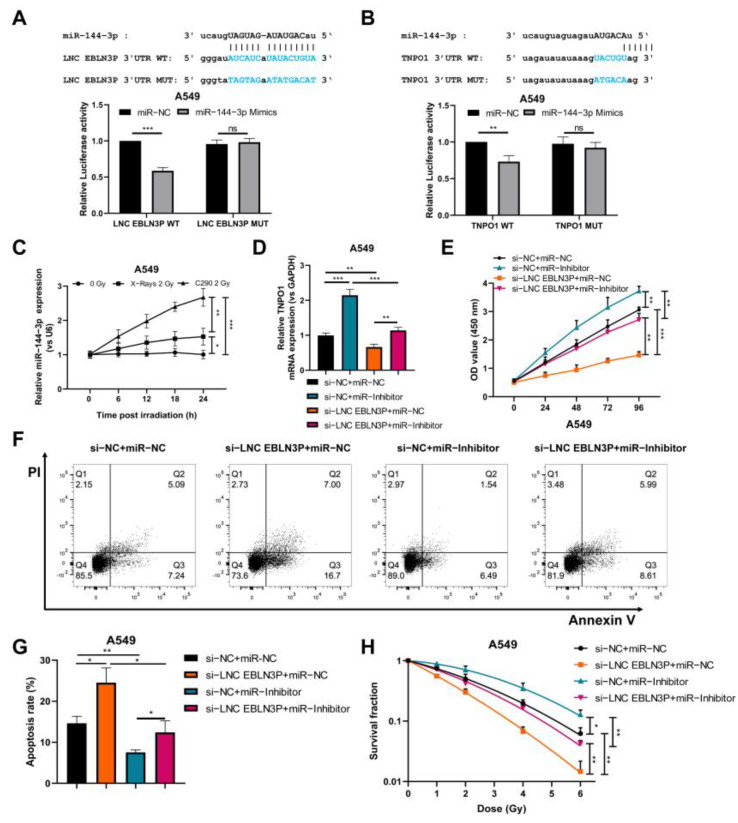
The regulation of LNC EBLN3P on TNPO1 is mediated by miR−144−3p. (**A**) A schematic diagram of the interaction sites between LNC EBLN3P and miR−144−3p. The relative luciferase activity of A549 cells was examined after co−transfection of LNC EBLN3P wild−type (WT) or mutant (MUT) luciferase reporter plasmids and miR−144−3p mimic or negative control (miR−NC). (**B**) The interaction between miR−144−3p and TNPO1 was examined by luciferase reporter assay. (**C**) The expression of miR−144−3p was upregulated over time in A549 cells irradiated with X−rays or carbon ions. (**D**–**H**) A549 cells were co−transfected with si−LNC EBLN3P or negative control (si−NC) and miR−144−3p inhibitor (miR−inhibitor) or negative control (miR−NC), and TNPO1 mRNA expression, cell viability, apoptosis, and radiosensitivity were examined. Data shown in (**A**–**H**) represent the mean ± SD (*n* = 3). * *p* < 0.05, ** *p* < 0.01, *** *p* < 0.001.

**Figure 6 cancers-15-00511-f006:**
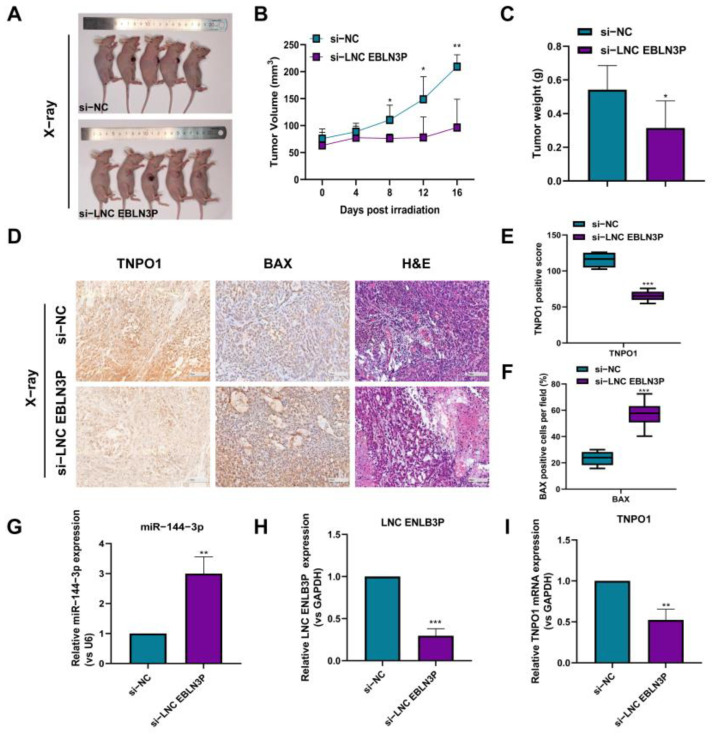
LNC EBLN3P silencing has the potential for radiosensitization of lung tumors. (**A**) Images of subcutaneous tumor tissues at the end of the treatment are shown. (**B**) Tumor volume changes with time. (**C**) The weights of dissected tumors are shown. (**D**) The tumor sections were subjected to hematoxylin–eosin staining and immunohistochemistry. Scale bar = 100 μm. (**E**,**F**) The TNPO1 and BAX protein levels in tumor tissues (panel D) were analyzed with Image−J software. (**G**–**I**) The relative expression levels of miR−144−3p, LNC EBLN3P, and TNPO1 in tumor tissues was examined by qRT−PCR. Data shown in (**A**–**I**) represent the mean ± SD (*n* = 3). * *p* < 0.05, ** *p* < 0.01, *** *p* < 0.001.

**Table 1 cancers-15-00511-t001:** Oligonucleotide siRNA sequences.

Gene	Sequence
si-NC sense si-NC antisense	5′-AGGUGUUGGUUAAUUGGUUUA-3′ 5′-AACACCUACAAACUAUACCUA-3′
si-LNC EBLN3P sense si-LNC EBLN3P antisense	5′-ACAGAAAGCACCAAACAGGGA-3′ 5′-CCUGUUUGGUGCUUUCUGUUU-3′
si-TNPO1 sense si-TNPO1 antisense	5′- GGAUUGUUACAAUAUUAUACU-3′ 5′- UAUAAUAUUGUAACAAUCCUA-3′

**Table 2 cancers-15-00511-t002:** Oligonucleotide primer sequences used for PCR.

Gene	Sequence
LNC EBLN3P forward LNC EBLN3P reverse	5′-TACGCGTTTTGGTCCCTGTT-3′ 5′-GCCACTTGGCTCAAAAGACTG-3′
miR-144-3p forward miR-144-3p reverse	5ʹ-GCCCCTACAGTATAGATGATGTA-3′ 5ʹ-GGATGCAGGTGCTGGAGGT-3′
U6 forward U6 reverse	5ʹ-CTCGCTTCGGCAGCACA-3′ 5ʹ-AACGCTTCACGAATTTGCGT-3′
GAPDH forward GAPDH reverse	5′-AGCCACATCGCTCAGACAC-3′ 5′-GCCCAATACGACCAAATCC-3′
TNPO1 forward TNPO1 reverse	5′-GTCTTAACAGAGTTAGAACTTGGG-3′ 5′-CTTCTGGGAGTATCTTGAAAGAG-3′

## Data Availability

The data presented in this study are available in this article (and Appendix A).

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
