# Peer review of "Heavy Ion-Responsive lncRNA EBLN3P Functions in the Radiosensitization of Non-Small Cell Lung Cancer Cells Mediated by TNPO1"

_cancers, 2023, doi:10.3390/cancers15020511_

Round 1

Reviewer 1 Report

First of all, the authors did remarkable work justifying the potential mechanism for two different radiation therapy on NSCLC. However, I believe there is still much space for the authors to improve in English and science.

In the English part, some of the sentences do not make sense or do not accurately explain what the authors want to describe.

For example:

Line 25: “In recent decades of years, the high-speed development of radiotherapy dramatically in-25 creased the cure rate of malignant tumors” need to be changed to “In the past decades, the rapid development of radiotherapy dramatically increased the survival rate of malignant tumors”. In addition, this also needs a reference to support this point of view.

Line 53 “the National Institute of Radiological Science (NIRS) in Japan”

Change to “The scholar in National Institute of Radiological Science (NIRS) of Japan”.

Line 57, “molecularmechanismsunderlyingtumor” need spacing

Line66 “There is mounting evidence”, 

increasing  is better for understanding

Line 68-69  “Our previous study found that” need reference

Line 195 “lncRNA EBLN3P was shown to play a significant role by interacting with miR-144-3p 195 and TNPO1 (Figure. 1A)” I can not see only from figure 1 A, which can support this point of view. The author needs to provide more data or enrich the mechanism diagram of figure 1A. You can use the diagram in Nature Reviews Molecular Cell Biology volume 22, pages 266–282 (2021), for reference.

Line 205 “correlation ship” typing mistake

Line 221,” and four candidate” the ”candidate“ is needless.

Line 325, “LNC EBLN3P knockdown significantly suppressed tumor development further.”  “further” is needless. 

Line 345-347 ” Authors should discuss the results and how they can be interpreted from the per perspective of previous studies and of the working hypotheses. The findings and their im-346 plications should be discussed in the broadest context possible. Future research directions may also be highlighted.“

The entire paragraph seems copied from somewhere else by mistake.

One science question, in the in-vivo study, why did the author only use X-ray, not the carbon ion beam?

Reviewer 2 Report

In the tesent study, the author explored the function of heavy ion-responsive lncRNA EBLN3P in radiosensitization of non-small cell lung cancer cells mediated by TNPO1.

For Results 3.5: Please also explore the expression of MiR-144-3p in respond to irradiation.

Reviewer 3 Report

Authors investigated the role of lncRNA EBLN3P in the treatment with heavy-ion radiotherapy in non-small cell lung cancer cells. They revealed that lncRNA EBLN3P is deeply involved in the apoptosis of A549 cells during heavy-ion radiotherapy. And LNC EBLN3P/miR-144-3p/TNPO1 axis plays a role in radiosensitization in vivo.

It seems like that EBLN3P regulates TNPO1, and miR-144-3p directly regulates both EBLN3P and TNPO1 based on the results showed by authors. To further prove that the role of EBLN3P/miR-144-3p/TNPO1 axis, author should perform experiments about knockdown of TNPON1 in A549 cells. Then repeat experiments in Figure 3 for TNPON1-knockdown A549 cells.

There are some typos in the text, like in figure legend of figure 3: ‘Dwonregulation’ of LNC EBLN3P inhibited the proliferation, cell colony formation and 266 increased the A549 cells apoptosis rate. It should be ‘Downregulation’ instead.

Round 2

Reviewer 1 Report

The answer fo the author is good to me. I agree to publish as present.

Reviewer 3 Report

I agree to accept the manuscript at its current form.